# Classification and Uncertainty Quantification of Corrupted Data using Semi-Supervised Autoencoders

## Abstract

Parametric and non-parametric classifiers often have to deal with real-world data, where corruptions like noise, occlusions, and blur are unavoidable – posing significant challenges. We present a probabilistic approach to classify strongly corrupted data and quantify uncertainty, despite the model only having been trained with uncorrupted data. A semi-supervised autoencoder trained on uncorrupted data is the underlying architecture. We use the decoding part as a generative model for realistic data and extend it by convolutions, masking, and additive Gaussian noise to describe imperfections. This constitutes a statistical inference task in terms of the optimal latent space activations of the underlying uncorrupted datum. We solve this problem approximately with Metric Gaussian Variational Inference (MGVI). The supervision of the autoencoder's latent space allows us to classify corrupted data directly under uncertainty with the statistically inferred latent space activations. Furthermore, we demonstrate that the model uncertainty strongly depends on whether the classification is correct or wrong, setting a basis for a statistical "lie detector" of the classification. Independent of that, we show that the generative model can optimally restore the uncorrupted datum by decoding the inferred latent space activations.

## 1 Introduction and Motivation

Many real-world applications of data-driven classifiers, e.g., neural networks, involve corruptions that pose significant challenges to the pretrained classifiers. Often, the corruption must previously be included, and thus already be known, during the process of training. For instance, noise (e.g., due to sensor imperfections) and convolutions (e.g., due to lens flares or unfocused images) are inevitable in image processing systems and may occur spontaneously and irregularly. The same holds for masking, which may occur when a foreign object occludes the actual object of interest (e.g., water droplets or dirt or scratches on the camera lens).

Hence, we aim to answer the following question in this paper: How can we classify corrupted data with a parametric classifier trained exclusively on uncorrupted data? As classifying corrupted data naturally demands a measure of uncertainty for validation (corruption may, in the worst case, lead to a total loss of information), we include both model uncertainty $\delta_m$ and reconstruction uncertainty $\delta_r$ in the classification. We refer to $\delta_m$ as the model's confidence on the classification itself. In contrast, we refer to $\delta_r$ as the confidence of the process on reconstructing the latent space activations given some corrupted datum. An overview of the proposed method is illustrated in Figure 1.

## 2 Classification and Uncertainty Quantification of Corrupted Data

### 2.1 Methodology Overview and Related Work

To address the challenge of classification and uncertainty quantification of corrupted data, we propose the following core approach, illustrated in Figure 2.

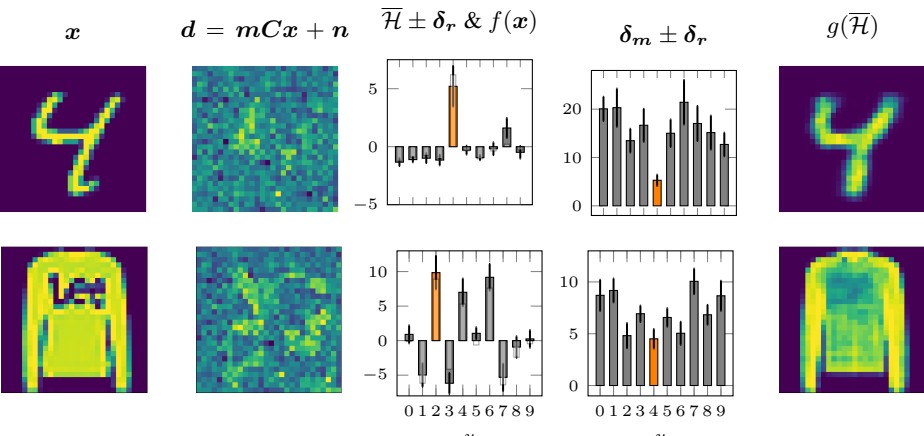

Figure 1: From left to right: Ground truth image $\boldsymbol{x}$ in the data space, corrupted image $\boldsymbol{d}$ in the data space (random masking $\boldsymbol{m}$, Gaussian blur $\boldsymbol{C}$, additive white Gaussian noise $\boldsymbol{n}$), posterior mean $\overline{\mathcal{H}}$ in the latent space with reconstruction uncertainty $\boldsymbol{\delta_r}$, model uncertainty $\boldsymbol{\delta_m}$ and the restored image $g(\overline{\mathcal{H}})$ (decoded posterior mean) in the data space. We have included the encoding of the uncorrupted data $f(\boldsymbol{x})$ (illustrated by the shaded white bars in the third column). Top row: data sample from the MNIST-dataset (ground truth label: 4). Bottom row: data sample from the Fashion-MNIST-dataset (ground truth label: 2 (pullover)). We can classify $\boldsymbol{d}$ using the posterior mean $\overline{\mathcal{H}}$ as the autoencoder's latent space is supervised (note the highlighted max. activation responsible for classification). We are able to classify and quantify model uncertainty $\boldsymbol{\delta_m}$ with the Mahalanobis-distance in the latent space (note the highlighted min. activation responsible for classification). Strong overlapping for the Fashion-MNIST-example of the $1 \cdot \sigma$ error bars of $\boldsymbol{\delta_r}$ across different classes indicates that no reliable and confident classification is possible due to heavy corruption.

①  **In the first step**, we train a semi-supervised autoencoder (Le et al., 2018) that is: (a) capable of classifying the input data with its latent space activations $\boldsymbol{h}$, and (b) capable of decoding the (supervised) latent space activations to generate higher-dimensional data, targeting it to be identical to the input. Except for these two constraints (a) and (b), we do not impose any further restrictions on the autoencoder and train it as a standard feedforward neural network.

②  **In the second step**, we decouple the decoder $g$ from the autoencoder and treat the decoder as a fixed generative function $g$. Neither retraining nor further modifying of $g$ is done in the following steps.

③  **In the third step**, we include $g$ in an ADDITIVE WHITE GAUSSIAN NOISE (AWGN) channel-model $\boldsymbol{d} = \boldsymbol{m}\boldsymbol{C}g(\boldsymbol{h}) + \boldsymbol{n}$. This AWGN channel model additionally involves heavy corruption like convolution $\boldsymbol{C}$ and masking $\boldsymbol{m}$.

④  **In the final step**, we approximate the posterior probability distribution $\mathcal{P}(\boldsymbol{h}|\boldsymbol{d})$ in the latent space, and derive the mean and standard deviation, corresponding optimally to some uncorrupted datum $g(\boldsymbol{h})$, given the corrupted datum $\boldsymbol{d}$. Due to supervision at the latent space, this reconstruction enables a direct classification of $\boldsymbol{d}$ including model and reconstruction uncertainty quantification, even though the decoding function was trained on uncorrupted data.

We use a set of samples $\mathcal{H}$ from the approximate posterior probability distribution to determine the sampling mean $\text{mean}(\mathcal{H}) = \overline{\mathcal{H}}$ as well as the set's reconstruction uncertainty $\boldsymbol{\delta_r}$ with the sampling standard deviation $\text{std}(\mathcal{H})$. Samples are statistically inferred by METRIC GAUSSIAN VARIATIONAL INFERENCE (MGVI) (Knollmüller & Enßlin, 2020).

In addition to reconstruction uncertainty $\boldsymbol{\delta_r}$, we determine the model uncertainty by calculating the MAHALANOBIS-distance (M-distance) (De Maesschalck et al., 2000) in the latent space representation, slightly different to Lee et al. (2018). For details of our implementation, see Algorithm 2 in the appendix. We here distinguish between reconstruction uncertainty $\boldsymbol{\delta_r}$ and model uncertainty $\boldsymbol{\delta_m}$ to evaluate the confidence of the process of inferring $\boldsymbol{h}$ and to evaluate the confidence of the classification given by the supervised latent space, respectively.

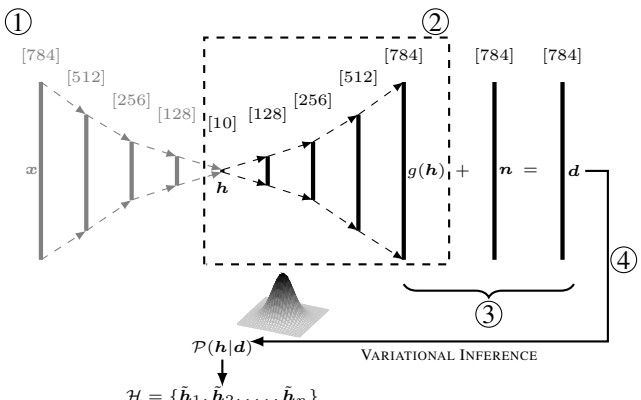

Figure 2: Concept visualization: Steps involved in classifying corrupted data and quantifying uncertainty of the reconstruction, as described in Section 2.1. The arrow from $\boldsymbol{d}$ to $\mathcal{P}(\boldsymbol{h}|\boldsymbol{d})$ graphically illustrates the process of statistically inferring the latent space activations from corrupted data $\boldsymbol{d}$ using MGVI. $\mathcal{H}$ includes all samples drawn from the posterior distribution in the latent space. Subsequently, the sampling mean and sampling standard deviation are determined from $\mathcal{H}$ to classify corrupted data samples and quantify their reconstruction uncertainty. Model uncertainty is determined via the Mahalanobis distance in the latent space. We do not depict convolution $\boldsymbol{C}$ and masking $\boldsymbol{m}$ in the figure above for simplicity.

Similarly to our approach, Böhm et al. (2019) and Böhm & Seljak (2020) have shown that the reconstruction of the latent space by posterior inference and by using generative models ((Adler & Öktem, 2018), (Seljak & Yu, 2019), (Wu et al., 2018)) for a corrupted datum can lead to an optimal image restoration with uncertainty quantification. These methods do, however, not focus on classifying the corrupted datum in the latent space, nor using supervised autoencoder structures.

In the field of quantifying uncertainties of classifications there exist several methods. Predominantly BAYESIAN NEURAL NETWORKS (BNNs) including neural network ensembling ((Depeweg et al., 2017), (Neal, 1995), (Pearce et al., 2020)) and MONTE CARLO-dropout (MC dropout) (Gal & Ghahramani, 2016) have lately shown success. More recently, EVIDENTIAL DEEP LEARNING (Sensoy et al., 2018) was introduced as yet another probabilistic method to quantify classification uncertainty. The latter two methods will be compared to our method in Section 3.

Finally, various methods to perform image restoration exist in the literature (see (Dong et al., 2012), (Lehtinen et al., 2018), (Mao et al., 2016a), (Mao et al., 2016b), (Zoran & Weiss, 2011)). Similar to the well-known denoising autoencoder (Vincent et al., 2008), almost all methods require prior knowledge of the corruption to be included in the training data. We argue that these methods lack flexibility, as they deal with one specific type of corruption. Once the model is trained, one cannot include other corruption types without retraining.

Moreover, many methods focus on either classification or eliminating corruption, but none of the named approaches combine both.

## 2.2 GENERATIVE MODEL AND BAYESIAN INFERENCE WITH NEURAL NETWORKS

The first step of our method is to train a semi-supervised autoencoder. The autoencoder involves the encoding function $f$ (mapping data $\boldsymbol{x} \in \mathbb{R}^p$ to the latent space representation with activations $\boldsymbol{h} \in \mathbb{R}^z, z \in \mathbb{N}$) as well as the decoding function $g$ (mapping $\boldsymbol{h}$ to the data space representation $\hat{\boldsymbol{x}} \in \mathbb{R}^p, p \in \mathbb{N}, p \gg z$). Parameters of $f : \mathbb{R}^p \to \mathbb{R}^z$ and $g : \mathbb{R}^z \to \mathbb{R}^p$ are optimized via a combination of two loss terms $\mathcal{L}_{gf}$ (representing reconstruction loss in the data space) and $\mathcal{L}_f$ (representing classification loss in the latent space):

$$\mathcal{L}_{\text{SAE}} = \mathcal{L}_{gf}(g(f(\boldsymbol{x})), \boldsymbol{x}) + \mathcal{L}_f(f(\boldsymbol{x})^j, \boldsymbol{y}) = \mathcal{L}_{gf}(\hat{\boldsymbol{x}}, \boldsymbol{x}) + \mathcal{L}_f(\boldsymbol{h}^j, \boldsymbol{y}). \qquad (1)$$

where $j$ denotes the number of dimensions of $\boldsymbol{h}$ that are supervised, i.e., $\boldsymbol{h}^j = [\boldsymbol{h}_1, \ldots, \boldsymbol{h}_j]$. The number of classes to be classified equals $j$. After normalizing all data samples (i.e., pixel values range

in between $[0, 1]$), we use the corresponding cross-entropy for each respective loss term to penalize false classifications in the latent space and inaccurate reconstructions in the data space. The binary cross-entropy represents the reconstruction loss, while we use the sparse categorical cross-entropy on integer labels $\boldsymbol{y} = [0, 1, \ldots, 9]$ to represent the classification loss. Note that for training, we process the latent space activations $\boldsymbol{h}$ through the softmax-function before feeding them to the sparse categorical cross-entropy. However, the softmax-function is not included as an activation function in our neural network. We minimize the general loss function Equation (1) using Adam optimizer (Kingma & Ba, 2015)[1].

Once the training procedure has converged, we decouple the decoding function $g$ from the autoencoder and extend it to model different types of corruption (this is necessary as the decoder is trained on uncorrupted data). Without loss of generality, we use an AWGN model including the nonlinearity $g(\boldsymbol{h})$, which additionally involves masking $\boldsymbol{m}$ and convolution $\boldsymbol{C}$ on $g$:

$$\boldsymbol{d} = \boldsymbol{m}\boldsymbol{C}g(\boldsymbol{h}) + \boldsymbol{n}. \tag{2}$$

In the data space, additive white Gaussian noise, $\boldsymbol{n} \in \mathbb{R}^p \sim \mathcal{N}(\mathbb{0}, \boldsymbol{\Sigma}_n)$, is applied to the decoded latent space signal $g(\boldsymbol{h})$, which yields the corrupted data $\boldsymbol{d} \in \mathbb{R}^p$. Note for the implementation of $\boldsymbol{h} = \boldsymbol{A}\boldsymbol{\xi} + \boldsymbol{\mu}_h$, the reparametrization trick Kingma & Welling (2014) is applied.[2]

In addition to AWGN, we include corruptions of masking $\boldsymbol{m}$ and convolutions $\boldsymbol{C}$, which are both linear operations. See the Appendix A, for details on the implementation of $\boldsymbol{n}, \boldsymbol{m}$ and $\boldsymbol{C}$.

Since we are interested in reconstructing the latent space activation $\boldsymbol{h}$ from $\boldsymbol{d}$ alongside uncertainty quantification, the goal is to determine the posterior distribution $\mathcal{P}(\boldsymbol{h}|\boldsymbol{d}) \propto \mathcal{P}(\boldsymbol{d}|\boldsymbol{h})\mathcal{P}(\boldsymbol{h})$. The log-probability distribution reads

$$-\ln \mathcal{P}(\boldsymbol{h}|\boldsymbol{d}) = \frac{1}{2}\left(\left(\boldsymbol{d} - \boldsymbol{m}\boldsymbol{C}g(\boldsymbol{h})\right)^{\mathrm{T}}\boldsymbol{\Sigma}_n^{-1}\left(\boldsymbol{d} - \boldsymbol{m}\boldsymbol{C}g(\boldsymbol{h})\right) + \left(\boldsymbol{h}^{\mathrm{T}}\boldsymbol{\Sigma}_h^{-1}\boldsymbol{h}\right)\right) + \text{const.}, \tag{3}$$

where $(\cdot)^{\mathrm{T}}$ denotes the matrix transpose. Since we are finally interested in the analytically intractable mean of $\boldsymbol{h}$, $\langle \boldsymbol{h} \rangle_{\mathcal{P}(\boldsymbol{h}|\boldsymbol{d})} = \int \boldsymbol{h}\mathcal{P}(\boldsymbol{h}|\boldsymbol{d})d\boldsymbol{h}$, we approximately determine mean and variance of $\mathcal{P}(\boldsymbol{h}|\boldsymbol{d})$ by applying MGVI. Similar to other variational inference methods (Kingma & Welling (2014), Kucukelbir et al. (2017)), MGVI approximates the distribution by a simper, but tractable distribution from within a variational family, $\mathcal{Q}(\boldsymbol{h})$. The parameters of $\mathcal{Q}(\boldsymbol{h})$, i.e., mean $\eta$ and covariance $\Delta$, are obtained by the minimization of the variational lower bound. The size of a full variational covariance scales quadratically with the number of latent variables. Taking these limitations into account, we employ MGVI, which locally approximates the target distribution using the inverse Fisher metric as an uncertainty estimate around the variational mean $\eta$, for which we optimize. The approximation is represented by an ensemble of samples $\mathcal{H} = \{\tilde{\boldsymbol{h}}_1, \tilde{\boldsymbol{h}}_2, \ldots, \tilde{\boldsymbol{h}}_n\}$ with $\tilde{\boldsymbol{h}}_n \in \mathbb{R}^z$, which we use for our analysis. $\tilde{(\cdot)}$ refers to the inferred sample. We here call $\overline{\mathcal{H}}$ the posterior mean and $\boldsymbol{\delta}_r$ the posterior standard deviation, or, the reconstruction uncertainty.

### 2.3 CLASSIFICATION AND UNCERTAINTY QUANTIFICATION

The supervision of the latent space allows us to classify the input $\boldsymbol{d}$ in a straightforward manner by evaluating the sampling mean and sampling standard deviation of the set $\mathcal{H}$. While the sampling mean of the set $\text{mean}(\mathcal{H}) = \overline{\mathcal{H}}$ gives the class of the most likely classification, the sampling standard deviation reflects the reconstruction uncertainty $\boldsymbol{\delta}_r$ of the latent space posterior distribution. $\boldsymbol{\delta}_r$ depends on the type and magnitude of the corruption as well as the prior probability distribution we include in the channel model (Equation (3)). We visualize this dependency with various experiments, see Figure 4 and Figure 10.

The straightforward classification does not yet provide information about the model uncertainty on the classification. Since we are additionally interested in the uncertainty of the model, $\boldsymbol{\delta}_m$, we evaluate the M-distance of all samples in $\mathcal{H}$ to all class conditional distributions in the latent space. The closest class conditional distribution to a single sample $\tilde{\boldsymbol{h}}_n$ corresponds to the most likely class. The absolute value of the M-distance to the closest class conditional distribution serves as a measure of the

---

[1] Test accuracy of $[98, 6\%; 89, 4\%]$ on the encoding function $f$ with [MNIST; Fashion-MNIST].

[2] $\boldsymbol{\Sigma}_h = \text{cov}(f(\boldsymbol{X}_{\text{Val}})), \boldsymbol{\Sigma}_h = \boldsymbol{A}\boldsymbol{A}^{\mathrm{T}}, \boldsymbol{\Sigma}_h \in \mathbb{R}^{z \times z}, \boldsymbol{\xi} \sim \mathcal{N}(\mathbb{0}, \boldsymbol{I}), \boldsymbol{\xi} \in \mathbb{R}^z, \boldsymbol{\mu}_h = \text{mean}(\boldsymbol{X}_{\text{Val}}), \boldsymbol{\mu}_h \in \mathbb{R}^z$

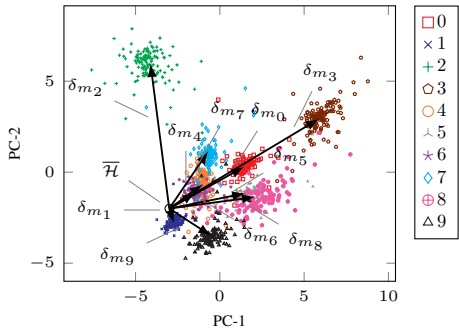

Figure 3: Illustration of the latent space structure of a semi-supervised autoencoder and the M-distance as a classifier based on MNIST. For this visualization, the 10-dimensional latent space activations are mapped to a two-dimensional subspace using a Principal Component (PC) analysis (Wold et al., 1987). For an arbitrary corrupted datum $d$, the inferred posterior mean $\overline{\mathcal{H}}$ in the latent space is marked accordingly. To classify $\overline{\mathcal{H}}$, the M-distance is computed to every cluster in the latent space to obtain $\boldsymbol{\delta}_{m_i}$ for all ten classes. The shortest distance $\arg\min(\boldsymbol{\delta}_m)$ serves as the classification. In this concrete example, the given posterior mean $\overline{\mathcal{H}}$ would likely be classified to digit 1. The absolute value of $\boldsymbol{\delta}_m$ reflects at the same time the model uncertainty w.r.t. each class in the latent space.

model uncertainty $\boldsymbol{\delta}_m$. In this work, all class conditional distributions in the latent space are assumed to follow multivariate Gaussian distributions with covariance $\boldsymbol{\Sigma}_i$ and mean $\boldsymbol{\mu}_i$. We determine the parameters of these class conditional distributions by passing the uncorrupted data samples from an independent (i.e., independent of training and testing) dataset $\boldsymbol{X}_{\text{Val}}$ (see Section 3) through the encoder $f$. This method is an implementation slightly different to Lee et al. (2018), where it was shown that the Mahalanobis metric is not only an accurate classifier in this context but also a reliable out-of-distribution detector reflecting the model uncertainty. Lee et al. (2018) uses tied covariance matrices instead of individual covariance matrices for each class conditional distribution, as done in our method.

Concretely, we calculate the M-distance of all samples in $\mathcal{H}$ to all class-conditional clusters $\mathcal{C}_k$ within the latent space, each characterized by $\boldsymbol{\mu}_{\mathcal{C}_k}$ and $\boldsymbol{\Sigma}_{\mathcal{C}_k}$ for $K$-classes. We then determine sampling mean $\overline{\boldsymbol{\delta}_m}$ and sampling standard deviation $\overline{\boldsymbol{\delta}_r}$ of the M-distances of all samples. This way, we can represent reconstruction uncertainty by the sampling standard deviation (resulting directly from the shape of the inferred posterior distribution) and model uncertainty by the absolute value of the M-distance. For a graphical illustration, refer to Figure 3. The pseudo-code is given in Algorithm 2 in the appendix.

## 2.4 SUMMARY AND LIMITATIONS

We summarize our proposed methodology (Algorithm 1) to classify a corrupted datum $d$ including uncertainty quantification, which requires the following input in addition to $d$:

$m, C$: Without loss of generality, here we assume corruption by masking and convolution represented by $m$ and $C$ in the AWGN channel-model, as written in Equation (2). The convolution $C$ can be determined with methods proposed by, e.g., Herbel et al. (2018), Hu & de Haan (2006) and Schlecht et al. (2006), occlusion $m$ can be modeled by, e.g., Li et al. (2013) and Rosales & Sclaroff (1998). $\boldsymbol{\Sigma}_n$: Noise covariance matrix. Noise is drawn from a Gaussian distribution with covariance $\boldsymbol{\Sigma}_n$ and mean $\boldsymbol{\mu}_n = \mathbb{0}$ and applied additively to the data $d$. Various methods exist to extract $\boldsymbol{\Sigma}_n$ given $d$ (e.g. (Gravel et al., 2004), (Liu et al., 2012), (Russo, 2003)). $\boldsymbol{\Sigma}_h$: Sampling covariance matrix of all (uncorrupted) latent space activations processed by the encoding function $f$. We use the assumption that an autoencoder can represent an inherent, sub-dimensional structure of the data in its latent space and assume this sub-dimensional structure to sufficiently follow a multivariate Gaussian probability distribution.

---

**Algorithm 1:** Classification and Uncertainty Quantification of Corrupted Data

---

**Input:** Decoder $g : \mathbb{R}^z \to \mathbb{R}^p$; corrupted datum $\boldsymbol{d}$; noise covariance $\boldsymbol{\Sigma}_n \in \mathbb{R}^{p \times p}$; corruption
models $\boldsymbol{m}$ and $\boldsymbol{C}$; latent space covariance $\boldsymbol{\Sigma}_h \in \mathbb{R}^{z \times z}$
**Output:** Classification $\hat{y}_d$, reconstruction uncertainty $\boldsymbol{\delta}_r$, model uncertainty $\boldsymbol{\delta}_m$; reconstruction
of uncorrupted datum $g(\overline{\mathcal{H}})$

1 Decouple decoder $g$ from semi-supervised autoencoder, pretrained on uncorrupted data
2 Define data model, $\boldsymbol{d} = \boldsymbol{m}\boldsymbol{C}g(\boldsymbol{h}) + \boldsymbol{n}$
3 Approximate $\mathcal{P}(\boldsymbol{h}|\boldsymbol{d})$ with MGVI and store samples in $\mathcal{H} = \{\tilde{\boldsymbol{h}}_1, \tilde{\boldsymbol{h}}_2, \ldots, \tilde{\boldsymbol{h}}_n\}$
4 Determine the sampling mean of $\mathcal{H}$ (i.e., $\overline{\mathcal{H}}$) and classify datum directly or via M-distance to
obtain $\hat{y}_d$
5 Quantify reconstruction uncertainty with sampling standard deviation from $\mathcal{H}$
6 Quantify model uncertainty using M-distance in the latent space
7 Generate reconstruction of uncorrupted datum $g(\overline{\mathcal{H}})$

---

## 3 Experiments

To experimentally validate our method of classifying corrupted data with a supervised autoencoder
trained on uncorrupted data, we conduct several experiments[3] on the MNIST (LeCun, 1998) and the
Fashion-MNIST (Xiao et al., 2017) dataset (both MIT-licenses, `https://opensource.org/licenses/MIT`). We evaluate the performance on various corruptions types and magnitudes and
perform a comparison to MC dropout (Gal & Ghahramani, 2016) and EDL (Sensoy et al., 2018).
The following architecture is used for the supervised autoencoder (we use the same architecture for
both datasets): A feedforward neural network is built with dimensions $784^{\{0\}} - 512^{\{1\}} - 256^{\{2\}} -
128^{\{3\}} - 10^{\{4\}} - 128^{\{5\}} - 256^{\{6\}} - 512^{\{7\}} - 784^{\{8\}}$, where layers $\{0\} - \{2\}$ and $\{4\} - \{7\}$
use the SeLU activation function (Klambauer et al., 2017), layer $\{3\}$ linear and layer $\{8\}$ sigmoid
activations. Note that in our case, for simplicity, the number of latent space dimensions $z$ is equal to
the number of supervised classes $j$, although $j \leq z$ holds generally. For experiments, we train the
neural network with the architecture above on two different datasets, MNIST and Fashion-MNIST.
We split each dataset into three subsets, $\boldsymbol{X}_{\text{Train}}$ ($48 \cdot 10^3$ samples, used for training), $\boldsymbol{X}_{\text{Test}}$ ($10 \cdot 10^3$
samples, used for testing and experiments) and $\boldsymbol{X}_{\text{Val}}$ ($12 \cdot 10^3$ samples, used for determining $\boldsymbol{\Sigma}_h$
and $\boldsymbol{\Sigma}_{\mathcal{C}_k} \ldots \boldsymbol{\Sigma}_{\mathcal{C}_K}$). We use Tensorflow-Keras (Chollet et al., 2015) (Apache License, version 2.0,
`http://www.apache.org/licenses/LICENSE-2.0`) to implement the neural networks
and the MGVI implementation of NIFTy (Selig et al., 2013) (General Public License, version 3.0,
`https://www.gnu.org/licenses/gpl-3.0.en.html`) to perform the inference.

### 3.1 Classification

We visualize experiments (1) – (3) in Figure 4. In the first experiment (1), we classify data from
an independent test set of the MNIST-dataset corrupted by different noise levels with the proposed
method. We compare the accuracy of our method to the baseline of processing corrupted data through
the encoder of the pretrained autoencoder. We show that we significantly improve the accuracy of
classifying corrupted data in comparison to the straightforward classification by $f(\boldsymbol{d})$. For the second
experiment (2), we use the same data samples as for (1) with the exception that we now additionally
corrupt the data with window masking (see Appendix for visualization and details) at a constant noise
level of $\alpha = 0.1$. Again, we compare the accuracy of our method to the baseline of processing the
same data samples through the encoder.

In the third experiment (3), we corrupt the data by convolving them with a Gaussian blur kernel with
a filter size of $7 \times 7$ and different magnitudes $\gamma$ at a constant noise level of $\alpha = 0.1$.

Experiments (1), (2), and (3) lead to the following conclusions:

- The reconstruction uncertainty $\boldsymbol{\delta}_{r_{\text{True}}}$ of correct classifications is approximately equivalent
to $\boldsymbol{\delta}_{r_{\text{False}}}$ of wrong classifications. It is thus independent of the classification.

---

[3]Code to be found ⭕ here.

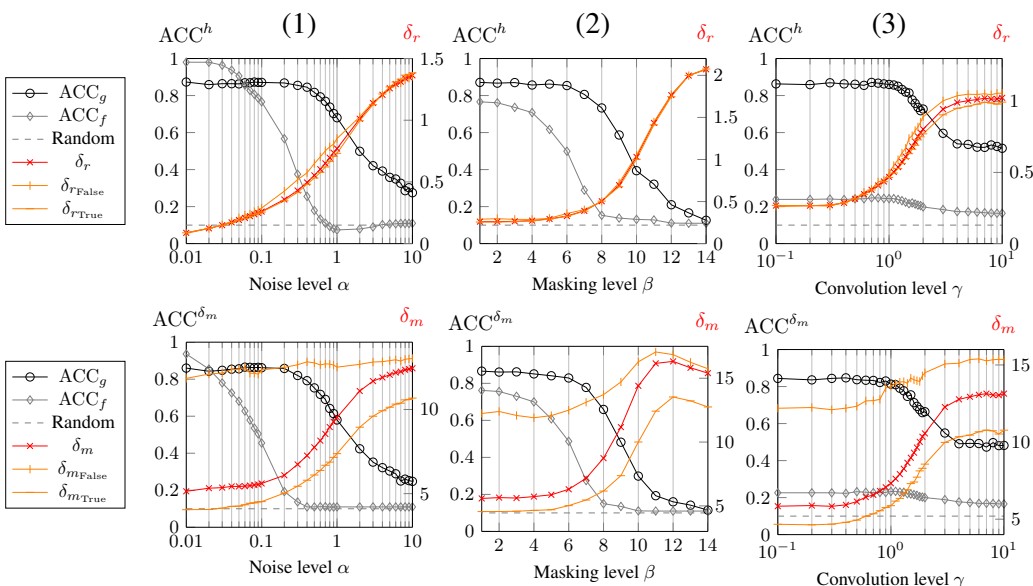

Figure 4: Accuracy and uncertainty (reconstruction uncertainty $\delta_r$ and model uncertainty $\delta_m$) of classifications of data samples of the MNIST dataset (for Fashion-MNIST see Figure 10 in the Appendix A) at different noise levels (left column), different masking levels (middle column), and different convolution levels (right column) exploiting the supervised latent space structure (top row) and the M-distance (bottom row) as classifying features.

$\text{ACC}_f$ serves as the baseline and is the accuracy of the plain encoding function $f$ classifying corrupted data. $\text{ACC}_g$ corresponds to the accuracy of the method proposed in Section 2.1. Additionally, we distinguish between uncertainties of correct classifications $\delta_{r_{\text{True}}}$, $\delta_{m_{\text{True}}}$ and of false classifications $\delta_{r_{\text{False}}}$, $\delta_{m_{\text{False}}}$.

The M-distance and thus $\delta_m$ strongly depends on the quality of the classification; $\delta_r$ depends on the the level of corruption and is independent of the classification result. The plot was generated with 1000 test samples for each data point (accordingly corrupted). The standard deviation is averaged over all 1000 samples.

- Opposed to $\delta_r$, the model uncertainty $\delta_m$ strongly depends on the correct/wrong classification of the corrupted datum: $\delta_m$ is significantly and consistently higher for false classifications than for true classifications. This delta in the model uncertainty corresponds to our method's concept of capturing the class conditional distribution density with the M-distance: For wrong classifications, the M-distance is larger because the corrupted sample is dissimilar to samples from its underlying class conditional distribution. This feature sets the basis for a statistical "lie detector" (see section 3.2) of classification and thus allows the possibility of various applications: One could pass data of high reconstruction uncertainty to further processing methods, as the process of inference is in this case likely faulty. Moreover, it is possible to pass detected false classifications with high model uncertainties to human verification. Fields of application could be the validation of neural networks in, e.g., medical imaging and other safety-critical applications.

- Classifying corrupted data through the decoder (rather than the encoder) with a suitable channel model considering the corruption significantly improves the model's accuracy without the necessity of retraining the autoencoder. Only for very low noise levels, the direct processing of the corrupted data through $f$ performs better than our method. We observe a slight loss of accuracy in the process of inference.

- Both uncertainties $\delta_r$ and $\delta_m$ rise with increasing levels of corruption, as expected. Also, the absolute value of the reconstruction uncertainty $\delta_r$ correlates inversely with the corresponding accuracy across all three experiments.

## 3.2 DETECTION OF FALSE CLASSIFICATIONS

Finally, in the experiment (4) (see Figure 5), motivated by the results of experiments so far, we validate the model uncertainty of our method by introducing the uncertainty based Receiver Operating Characteristics (U-ROC) curve of detecting false classifications with the M-distance. We evaluate the binary classification task of the two classes "*The neural network correctly classifies a corrupted datum*" (POSITIVE CLASS) and "*The neural network falsely classifies a corrupted datum*" (NEGATIVE CLASS). Based on the model uncertainty of our method, we aim to predict the two classes without further knowledge, providing the initially proposed "lie detector". The U-ROC curve is built from the TRUE POSITIVE RATE and the FALSE POSITIVE RATE.

We detect a false classification if the minimum M-distance of a reconstructed sample in the latent space is above some threshold value. On the contrary, we detect a correct classification if the minimum M-distance is below the threshold. These threshold values vary for the plot depicted in Figure 5 between 0 and 12, being confident at 0 and uncertain at 12. We show by Figure 5 that the model uncertainty of our methodology truly reflects the confidence on a specific classification, verifying that high model uncertainty correlates with false classifications.

We compare our U-ROC curve with the U-ROC curve of the MC dropout method (Gal & Ghahramani, 2016) and with the U-ROC curve of EDL (Sensoy et al., 2018), feeding all methods with the identical input of a datum corrupted by noise at $\alpha = [0.1, 0.5, 1.0]$. Kindly note that this comparison uses the optimized neural network architectures presented in the respective publications, which is different from our simplistic proof-of-concept architecture: Both EDL and MC dropout use the LeNet (Lecun et al., 1998) with custom modifications[4,5], while we use a significantly simpler feedforward neural network, see Figure 2. Note that our method is not limited to feedforward neural networks and applicable to convolutional neural networks, as well. MC dropout exploits weight-dropout in a neural network to achieve statistically varying outputs of their classifying neural network at the same input over several forward passes. They argue that overlapping output samples indicate high uncertainty in the classification – we use the number of overlapping samples as the metric for detecting false classifications (applying 50 repetitions per sample). On the contrary, EDL trains the neural network to learn parameters of a Dirichlet distribution, instead of softmax probabilities. By replacing the standard output of a classification network (softmax) with the parameters of a Dirichlet density, EDL represents the predictions of the neural network as a distribution over possible softmax outputs, rather than the point estimate of a softmax output. The output of EDL equals the range of the possible entropies, i.e., $[0, \log(10)]$. To create the U-ROC curve of EDL, we thus use different thresholds ranging from 0 to 1.

We make the following conclusions from experiment (4), Figure 5:

- Our method seems to outperform MC dropout and EDL to detect false classifications given the same data samples at the input for $\alpha = 0.1$ and $\alpha = 0.5$. One reason for this might be that the M-distance serves as a reliable out-of-distribution detector, exploiting the inherent latent space structure of uncorrupted data as a reference, as opposed to MC dropout and EDL. For $\alpha = 1.0$, both EDL and our method outperform MC dropout, while the AUC of EDL is largest. Here, it should be noted that EDL cannot classify the corrupted data at this noise level (accuracy: 8.9%), resulting in only few samples to test the cases of TRUE POSITIVES and FALSE POSITIVES.

- All three methods provide reliable results for detecting false classifications for low noise levels.

- The model uncertainty $\delta_m$ truly reflects the confidence of the classification, i.e., a high value of $\delta_m$ correlates empirically with a higher probability of false classification.

- U-ROC curve combined with the accuracy indicates that EDL seems to overestimate uncertainties, leading to a very robust U-ROC curve for high noise levels, but simultaneously leading in presence of data corruption to a severe drop in the accuracy of the model.

---

[4]MC dropout: dropout is applied before the last fully connected inner-product layer
[5]EDL: LeNet trained to learn parameters of a Dirichlet distribution, instead of softmax probabilities

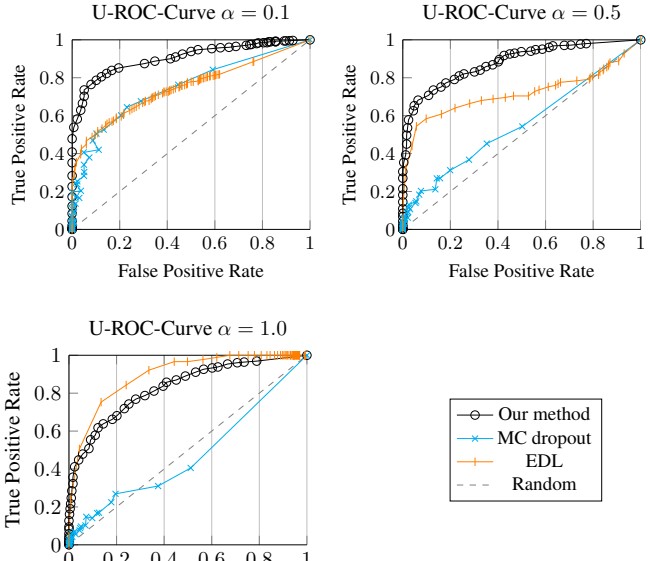

| $\alpha$ | METHOD | ACC | AUC |
|---|---|---|---|
| | Our Method | 0.862 | 0.904 |
| 0.1 | MC dropout | 0.915 | 0.745 |
| | EDL | 0.666 | 0.746 |
| | Our Method | 0.753 | 0.881 |
| 0.5 | MC dropout | 0.755 | 0.555 |
| | EDL | 0.125 | 0.717 |
| | Our Method | 0.580 | 0.828 |
| 1.0 | MC dropout | 0.570 | 0.472 |
| | EDL | 0.089 | 0.885 |

Figure 5: Uncertainty based Receiver Operator Characteristics (U-ROC) of the proposed identifier of false classifications for different noise levels $\alpha$ of our method in comparison with MC dropout and with EDL. In this experiment, the formulation of the e.g. "TRUE NEGATIVE" case would be: *Based on the uncertainty value, the sample is correctly detected as a false classification* – the "lie detector" works. Samples are taken from the MNIST-dataset. Top left: corrupted datum at $\alpha = 0.1$. Bottom left: corrupted datum at $\alpha = 1.0$. The irregularity in the U-ROC-Curve of the dropout model is due to the stochastic nature of MC dropout. Bottom right: Evaluation of Accuracy (ACC) for all given noise values $\alpha$ and the Area under the Curve (AUC) for all U-ROCs.

## 4 SUMMARY AND FUTURE RESEARCH

We present a novel approach to classify heavily corrupted data with parametric classifiers trained on uncorrupted data. As we build our procedure on a probabilistic architecture, we quantify both classification and model uncertainty, allowing for a reliable detection of false classifications. We see our method as a highly flexible template that can be applied to any generative neural network to improve performance on corrupted data significantly. If the generative neural network comes with a supervised encoded space, it can classify the data directly. We have shown that the M-distance can independently be used to classify data. Limitations of our method include that the corruption type needs to be modeled.

Future research is planned in more realistic, real-time, and complex scenarios with strong corruptions, e.g., medical imaging, autonomous driving, or astronomy. Here, potential applications of our method could be image segmentation and object detection via bounding boxes. The method described in Section 2.1 can provide uncertainties of the bounding box and thus of the position of the detected object. This might be useful, e.g., if the object of interest is occluded. A typical situation would be a car visually blocking pedestrians at a crossing.

## 5 ACKNOWLEDGEMENTS

Anonymous.

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

## A APPENDIX

### A.1 CORRUPTION LAYOUT

Here we visualize the effect of the different corruptions, i.e., convolution as Gaussian blur $C$, Figure 8, masking $m$ (Figure 7, Figure 9), and noise $n$, (Figure 6). We evaluate the effect of different corruption levels on accuracy and confidence, where

- $\alpha$ corresponds to the standard deviation of the Gaussian distribution from which noise is drawn, i.e., $\mathbf{\Sigma}_n = \mathbf{I} \cdot \alpha$ ($\mathbf{I} \in \mathbb{R}^{p \times p}$ is the identity matrix)
- $\beta$ corresponds to the number of columns and rows of pixels set to 0, counted from outside to inside (i.e., $\beta = 0$ means no masking, $\beta = 14$ means full image is masked), see figure Figure 7,
- $\gamma$ corresponds to the standard deviation of the Gaussian blur kernel with a filter size of $7 \times 7$ pixels.

Note that to the experiments conducted on the effect on convolution (Figure 4, right), we added noise at $\alpha = 0.1$. For visualization purposes, we do not add noise for Figure 8.

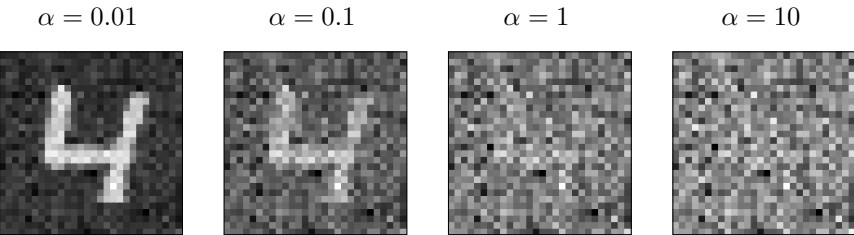

Figure 6: Exemplary visualization of corruption through noise. Experiments cover the entire noise range from $\alpha = 0.01$ to $\alpha = 10$.

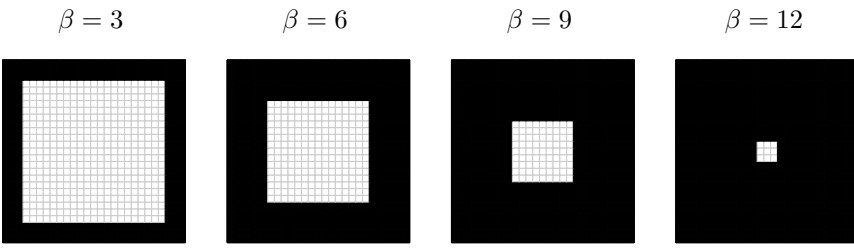

Figure 7: Exemplary visualization of isolated masking. Experiments cover the entire masking range from $\beta = 0$ to $\beta = 14$ and additional noise at $\alpha = 0.1$. The experiment layout of masking is adopted from

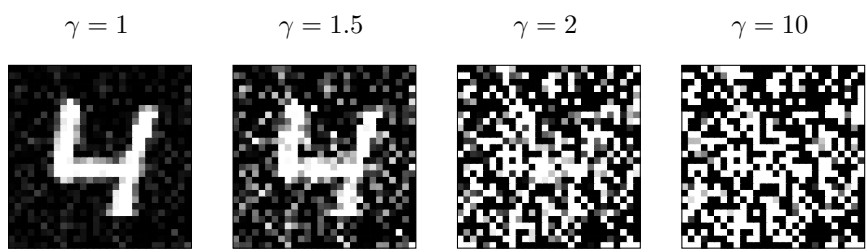

Figure 8: Exemplary visualization of corruption through convolution. Experiments cover the entire convolution range from $\gamma = 0.1$ to $\gamma = 10$.

$$\alpha = 0.1, \beta = 3 \qquad \alpha = 0.1, \beta = 6 \qquad \alpha = 0.1, \beta = 9 \qquad \alpha = 0.1, \beta = 12$$



Figure 9: Exemplary visualization of different masking levels applied in experiments with additive noise $\alpha = 0.1$ on top. Experiments cover the entire masking range from $\beta = 0$ to $\beta = 14$.

## A.2 DETERMINATION OF THE MAHALANOBIS-DISTANCE

In addition to the methodology outlined in Section 2.3, we present with Algorithm 2 the pseudo-code of calculating the M-distance of the inferred latent space samples $\mathcal{H}$ to determine the model uncertainty $\boldsymbol{\delta}_m$.

---

**Algorithm 2:** Classification and Model Uncertainty by Mahalanobis Distance

---

**Input:** Set of posterior samples $\mathcal{H} = \{\tilde{\boldsymbol{h}}_1, \ldots, \tilde{\boldsymbol{h}}_n\}$; $\boldsymbol{\mu}_k, \ldots, \boldsymbol{\mu}_K$; $\boldsymbol{\Sigma}_{\mathcal{C}_k}, \ldots, \boldsymbol{\Sigma}_{\mathcal{C}_K}$
**Output:** Label of Classified Class $y$, model uncertainty $\boldsymbol{\delta}_m$

1 **for** $n \leftarrow 0$ to $N$:
2     **for** $k \leftarrow 0$ to $K$:
3         $\boldsymbol{\delta}_m[k]_n = \sqrt{[\tilde{\boldsymbol{h}}_n - \boldsymbol{\mu}_k]\boldsymbol{\Sigma}_{\mathcal{C}_k}^{-1}[\tilde{\boldsymbol{h}}_n - \boldsymbol{\mu}_k]^{\mathrm{T}}}$
4     $\mathcal{D}\{n\} = \boldsymbol{\delta}_{m_n}$
5 $y = \mathrm{argmin}(\mathrm{mean}(\mathcal{D}))$
6 $\boldsymbol{\delta}_m = \mathrm{mean}(\mathcal{D})$
7 **return** $\boldsymbol{\delta}_m, y$

---

## A.3 Experiments on Fashion-MNIST Data

Finally, we show in Figure 10 the results of the identical experiment layout of Figure 4 with Fashion-MNIST data. Note that we trained the neural network with the identical architecture as for MNIST data. Besides the general loss in the accuracy due to the retraining of the identical neural network on the more complex Fashion-MNIST dataset, the results are mostly coherent to Figure 4. In column 2, row 1, we observe that the accuracy does not further decrease for $\beta = 8$ through $\beta = 10$. We assume that this is due to the nature of the Fashion-MNIST dataset, where the window mask does not take away much information for the affected pixels: Classes such as T-shirt/top, Pullover, Dress, and Coat all usually exhibit the same structure in the affected area of the image.

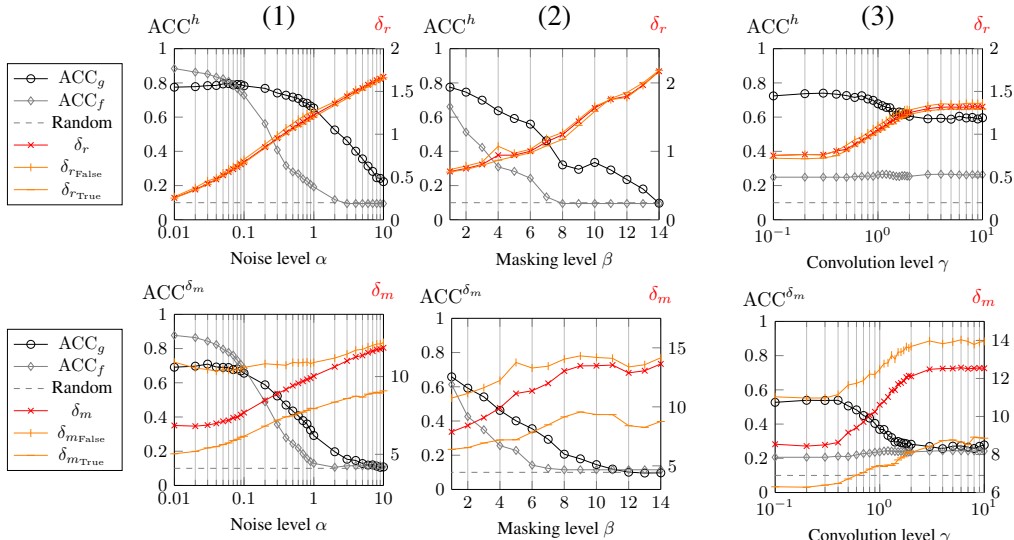

Figure 10: Accuracy and uncertainty (reconstruction uncertainty $\delta_r$ and model uncertainty $\delta_m$) of classifications of data samples of the Fashion-MNIST dataset (for MNIST see Figure 4) at different noise levels (left column), different masking levels (middle column), and different convolution levels (right column) exploiting the supervised latent space structure (top row) and the M-distance (bottom row) as classifying features.
$\text{ACC}_f$ serves as the baseline and is the accuracy of the plain encoding function $f$ classifying corrupted data. $\text{ACC}_g$ corresponds to the accuracy of the method proposed in Section 2.1. Additionally, we distinguish between uncertainties of correct classifications $\delta_{r_{\text{True}}}, \delta_{m_{\text{True}}}$ and of false classifications $\delta_{r_{\text{False}}}, \delta_{m_{\text{False}}}$.

## A.4 Experiment Layout for U-ROC curve

We use the following threshold range for the displayed methods:

- The range for U-ROC-Curve of Mahalanobis-Distance: $[0, 12]$
- The range for U-ROC-Curve of overlapping samples of MC dropout-model: $[0, 50]$
- The range for the U-ROC-Curve for EDL: $[0, 1]$.

Noise is applied at $\alpha = [0.1; 0.5; 1.0]$ without further corruption. The AUC is calculated using the trapezoidal rule. Implementation is inspired by Park (2019) for MC dropout and by Douglas (2021) for EDL.

