# OpenReview forum: "Classification and Uncertainty Quantification of Corrupted Data using Semi-Supervised Autoencoders"
_ICLR.cc/2022/Conference — ICLR 2022 Submitted_

### Official Review · Reviewer_hL2w · 2021-10-18

**Correctness:** 4
**Technical Novelty And Significance:** 2
**Empirical Novelty And Significance:** 2
**Recommendation:** 3
**Confidence:** 4

**Main Review:**

The major concern I have is about the applicability of the method, given the limited experiments provided. In Section 3.1, the proposed method is tested on fairly simple data sets (MNIST and Fashion MNIST) and the baseline method used also relies on the encoding function to classify the input samples. Since the proposed method implies a major modification to commonly used networks for classification (e.g., ResNet), I believe that the authors should make an effort to support the applicability of the proposed technology on more complex data sets, such as CIFAR100 and ImageNet. There, it will be interesting to compare the proposed method to the following baseline: (i) apply an off-the-shelf method for image restoration that also quantifies uncertainty (e.g., using a variant of [1] that also relies on VAE), and (ii) feed the recovered image to a standard classifier to predict the unknown label (can possibly be used to assess prediction uncertainty as well). As far as I understand, the authors estimate the parameters of the corruption model (noise covariance, $m$, and $C$), so it is possible to apply a restoration algorithm given this information.

[1] Edupuganti V, Mardani M, Vasanawala S, Pauly J. Uncertainty Quantification in Deep MRI Reconstruction. IEEE Trans Med Imaging. 2021 Jan;40(1):239-250.

In Section 3.2 (detecting false classification), there are various competitive techniques developed for selective classification, see, e.g., [2], which are not discussed at all. Please compare the performance of the proposed approach to such selective classification methods, and use more complex data sets than MNIST.

[2] Geifman, Yonatan, and Ran El-Yaniv. "Selectivenet: A deep neural network with an integrated reject option." International Conference on Machine Learning. PMLR, 2019.


**Summary Of The Paper:**

This paper presents a method that simultaneously classifies corrupted data and quantifies uncertainty, despite the model being fitted only on uncorrupted data. The idea is to fit a semi-supervised autoencoder. Then, the encoded representation $h$ is fed into a decoder $g$, and the output $g(h)$ is augmented with various perturbations (additive noise, blur, and masking), resulting in a corrupted image $d$. Lastly, the method of MGVI is used to estimate $P(h \mid d)$, which is then used both to classify $d$ and to estimate the uncertainty. In experiments, the authors show the advantage of this method over baseline methods.


**Summary Of The Review:**

The paper presents an interesting application of VAE to the task of image classification with corrupted data. The ability to quantify the uncertainty is important, however, in its current form, the paper only presents a proof-of-concept rather than a technology that can be deployed in real-world applications.

---

> ### Author Response · Authors · 2021-11-29
> **Thank you for your review**
>
> Thank you for your review – we appreciate the feedback.
>
> We acknowledge that further experiments with more realistic datasets and a comparison to other competitive algorithms would better justify our method. We will consider including your main concerns in future research and further experiments.

---

### Official Review · Reviewer_Hjos · 2021-11-02

**Correctness:** 1
**Technical Novelty And Significance:** 2
**Empirical Novelty And Significance:** 2
**Recommendation:** 5
**Confidence:** 4

**Main Review:**

Strengths:
1. This paper proposed a novel approach to classify heavily corrupted data with parametric classifiers trained on uncorrupted data.
2. The proposed method can quantify both classification and model uncertainty, allowing for reliable detection of false classifications.
3. The authors show that the M-distance can independently be used to classify data.

Weakness:
1. It lacks extensive empirical validation in real-world datasets.
2. Some important baselines are missing.
3. An ablation study is missing.


**Summary Of The Paper:**

This paper presents a novel approach to classify heavily corrupted data with parametric classifiers trained on uncorrupted data. The proposed method can quantify both classification and model uncertainty, allowing for reliable detection of false classifications.

**Summary Of The Review:**

As this paper did not conduct the theoretical analysis, and the proposed method is some combination of some existing methods, I would expect more extensive empirical analysis in the experiments.
1. Add more complex datasets, such as CIFAR10, CIFAR100. Note that MNIST and Fashion-MNIST are more like a toy dataset.
2. As Sec3.1 focuses on classification experiments, it is better to add some baselines related to corrupted data classification.
3. An ablation study is necessary since the proposed method is a combination of some existing methods.

Some other issues:
1. The motivation about the M-distance is not clear, why it can be used as model uncertainty to detect OOD.

---

> ### Author Response · Authors · 2021-11-29
> **Thank you for your review**
>
> Thank you for your review – we appreciate the feedback.
>
> We acknowledge that further experiments with more realistic datasets would better justify our method. We will consider including your main concerns in future research and further experiments.

---

### Official Review · Reviewer_kh1g · 2021-11-02

**Correctness:** 3
**Technical Novelty And Significance:** 2
**Empirical Novelty And Significance:** 1
**Recommendation:** 5
**Confidence:** 3

**Main Review:**

1) If we have access to the clean training data, another intuitive way is to augment the clean data with corrupted data. How does this heuristic method perform compared with the proposed method? Some references on data augmentation: [1] [2]

2) For the loss of semi-supervised autoencoder (formula 1), do the two losses have the same magnitude? Will it benefit us if we add a parameter to the trade-off between the reconstruction and classification?

3) Since the experiments in this paper focus on simple datasets (e.g., MNIST and Fashion-MNIST), so it is not clear whether the proposed method will be computationally efficient for a large-scale dataset such as ImageNet. Will it introduce significant computational overhead to approximate the posterior probability distribution and calculate the M-distance when latent code is in a much larger dimension?

4) Algorithm 1 is not clear: for example, the input includes the corrupted datum d, however, inside the algorithm, it will create the d again. It is confusing how this algorithm will work during test time when only corrupted data is given. Also, when the test data comes in an online matter (not in a batch), will this method still work?


Ref:
[1] Hendrycks, Dan, et al. "The many faces of robustness: A critical analysis of out-of-distribution generalization." Proceedings of the IEEE/CVF International Conference on Computer Vision. 2021.

[2] Hendrycks, Dan, et al. "Augmix: A simple data processing method to improve robustness and uncertainty." arXiv preprint arXiv:1912.02781 (2019).

**Summary Of The Paper:**

This paper proposes a framework that tries to classify corrupted data while using models trained merely on clean data. Additionally, this framework also quantifies the classification uncertainty by using the Mahalanobis-distance.

**Summary Of The Review:**

Given the concerns I have listed above, I suggest “5: marginally below the acceptance threshold”.

---

> ### Author Response · Authors · 2021-11-29
> **Thank you for your feedback**
>
> Thank you for your review – we appreciate the feedback. We would like to address your main concerns in the following lines:
>
> 1. Our aim is to point out that our method is capable of handling corrupted data even though the model itself is not trained in this domain. Thus, we decided to keep the training data clean.
>
> 2. The losses are roughly of the same magnitude and as good performance was obtained for both reconstruction and classification, we did not introduce a loss-term-weighting. Indeed, introducing a weighting factor could improve performance, especially for more complex data sets.
>
> 3. We acknowledge that further experiments need to be conducted on more complex datasets to justify our method.
>
> 4. $d$ is in fact not created in the algorithm -- it is only the data model that is set up, where the corrupted datum $d$ is defined by the previously described channel model. $d$ is approximated in the following steps of the algorithm by finding the best fit of the latent space activations $h$. The algorithm is capable of classifying a corrupted datum if the pre-trained decoder $g$ and the corruption model are given.
>
> Again, many thanks you for your review. We will consider including your main concerns in future research and further experiments.

---

### Official Review · Reviewer_SCcY · 2021-11-03

**Correctness:** 2
**Technical Novelty And Significance:** 2
**Empirical Novelty And Significance:** 2
**Recommendation:** 3
**Confidence:** 3

**Main Review:**

Pros,

This paper proposes a novel approach that classifies corrupted data.

Cons,

The proposed method is not justified well. The algorithm is a combination of existing methods. For the algorithm, the authors should discuss why each part should be there and what will happen when we replace each part with other techniques.

This paper is related to Bayesian deep learning, deep Bayesian learning, adversarial attack, adversarial training, calibration, among others. A more comprehensive literature survey is required.

The experiment section is weak. The experiments provide comparisons only to MC dropout (2016) and EDL (2018). There are more recent algorithms. Moreover, experiments only with MNIST with a single neural net structure are not enough to validate the proposed algorithm.

**Summary Of The Paper:**

This paper proposes a false classification detection method. The proposed method first trains an autoencoder using uncorrupted data to obtain a decoder to define a generative model. The generative model is then exploited for quantifying the uncertainty of the model. Using MGVI, the generative model infers the posterior distribution of the corresponding latent vector for each data. The latent space can explain the model uncertainty, which can be measured by the Mahalanobis distance. From experiments, it is shown that the proposed method can detect corrupted data more accurately compared to MC dropout and EDL for MNIST dataset.


**Summary Of The Review:**

Overall, this paper provides a heuristic algorithm without any theoretical justification. Then, the experiment section should be very strong to be published. However, the experiment section of this work is not good enough.

---

> ### Author Response · Authors · 2021-11-29
> **Thank you for your feedback**
>
> Thank you for your review – we appreciate the feedback.
>
> We acknowledge that further experiments with more realistic datasets and a comparison to more recent algorithms would better justify our method. We will consider including your main concerns in future research and further experiments, including other neural network structures as well.

---

### Decision · Program_Chairs · 2022-01-20

**Decision:**

Reject

**Comment:**

This paper introduces a method for classifying corrupted data and quantifying uncertainty by training semi-supervised autoencoders only on clean (uncorrupted) data.

Pro:  The approach is novel utilizing metric Gaussian variational inference.

Cons: More thorough experiments are needed:  (1) extensive experiments on more complex data, (2) ablation study, (3) comparison to additional baselines.

Summary:  The paper introduces a novel method, however experiments are limited.